# Early Alterations of RNA Binding Protein (RBP) Homeostasis and ER Stress-Mediated Autophagy Contributes to Progressive Retinal Degeneration in the *rd10* Mouse Model of Retinitis Pigmentosa (RP)

**DOI:** 10.3390/cells12071094

**Published:** 2023-04-06

**Authors:** Alfred Yamoah, Priyanka Tripathi, Haihong Guo, Leonie Scheve, Peter Walter, Sandra Johnen, Frank Müller, Joachim Weis, Anand Goswami

**Affiliations:** 1Institute of Neuropathology, University Hospital RWTH Aachen, 52074 Aachen, Germany; ayamoah@ukaachen.de (A.Y.); ptripathi@ukaachen.de (P.T.); hguo@ukaachen.de (H.G.); leonie.scheve@rwth-aachen.de (L.S.); 2EURON—European Graduate School of Neuroscience, Maastricht University, 6229 ER Maastricht, The Netherlands; 3Department of Ophthalmology, University Hospital RWTH Aachen, 52074 Aachen, Germany; pwalter@ukaachen.de (P.W.); sjohnen@ukaachen.de (S.J.); 4Institute of Biological Information Processing, Molecular and Cellular Physiology, IBI-1, Forschungszentrum Jülich GmbH, 52425 Jülich, Germany; f.mueller@fz-juelich.de; 5Department of Neurology, Center for Motor Neuron Biology and Disease, Columbia University, New York, NY 10032, USA; 6Department of Neurology, Eleanor and Lou Gehrig ALS Center, Columbia University, New York, NY 10032, USA

**Keywords:** *rd10* retina, ER-Ca^2+^ homeostasis, ER chaperones, autophagy, RBP aggregation, neurodegeneration

## Abstract

The retinal degeneration 10 (*rd10*) mouse model is widely used to study retinitis pigmentosa (RP) pathomechanisms. It offers a rather unique opportunity to study trans-neuronal degeneration because the cell populations in question are separated anatomically and the mutated *Pde6b* gene is selectively expressed in rod photoreceptors. We hypothesized that RNA binding protein (RBP) aggregation and abnormal autophagy might serve as early pathogenic events, damaging non-photoreceptor retinal cell types that are not primarily targeted by the *Pde6b* gene defect. We used a combination of immunohistochemistry (DAB, immunofluorescence), electron microscopy (EM), subcellular fractionation, and Western blot analysis on the retinal preparations obtained from both *rd10* and wild-type mice. We found early, robust increases in levels of the protective endoplasmic reticulum (ER) calcium (Ca^2+^) buffering chaperone Sigma receptor 1 (SigR1) together with other ER-Ca^2+^ buffering proteins in both photoreceptors and non-photoreceptor neuronal cells before any noticeable photoreceptor degeneration. In line with this, we found markedly altered expression of the autophagy proteins p62 and LC3, together with abnormal ER widening and large autophagic vacuoles as detected by EM. Interestingly, these changes were accompanied by early, prominent cytoplasmic and nuclear aggregation of the key RBPs including pTDP-43 and FET family RBPs and stress granule formation. We conclude that progressive neurodegeneration in the *rd10* mouse retina is associated with early disturbances of proteostasis and autophagy, along with abnormal cytoplasmic RBP aggregation.

## 1. Introduction

Retinitis pigmentosa (RP) is an inherited retinal disease resulting in the progressive degeneration of retinal photoreceptors and ultimately leading to blindness [1,2]. Mutations in more than 100 genes are known to cause RP [3,4] (also see the homepage of the retinal information network, http://www.sph.uth.tmc.edu/Retnet/ (accessed on 21 December 2022)). Mutations in genes involved in the phototransduction cascade, including opsins, transducin, cyclic GMP (cGMP) phosphodiesterase type 6 (Pde6), guanylate-cyclase-activating protein, and cGMP-gated channels, are major causes of retinal photoreceptor degeneration in human patients [5,6,7,8].

Several mouse models of RP have been described thus far [9,10,11,12]. The *rd10* mouse carries a spontaneous point mutation in the *Pde6b* gene and displays progressive photoreceptor degeneration starting at postnatal day *P17-18* with peak photoreceptor death at *P25*. This model has been widely used to study RP pathophysiology [11,12,13,14,15]. The *Pde6b* gene encodes rod phosphodiesterase (PDE), an enzyme that hydrolyzes cGMP to GMP in response to light. A point mutation in the *Pde6b* gene dramatically decreases maximal and basal PDE6 activity, apparently caused by a decrease in protein stability and reduced transport of the PDE6 protein to the outer segment of the retina [16]. The reduced availability of PDE6 results in reduced activity, which eventually leads to the accumulation of free cGMP in photoreceptor cells. This further leads to uncontrolled opening of the cGMP-gated channels, causing excessive Ca^2+^ influx [11,12,13,14,15,16]. Altered Ca^2+^ homeostasis can trigger photoreceptor degeneration involving excitotoxicity, endoplasmic reticulum (ER) stress, autophagy defects, inflammation, and apoptosis [17,18,19,20,21]. Interestingly this mutation produces no alteration of *Pde6b* RNA, and the enzymatic properties of the remaining mutant PDE6 appear to be nearly normal [11,12,13,14,15,16].

In many forms of RP, and especially in *rd10* mice carrying the *Pde6b* gene mutation, the disease is characterized by the degeneration of rods followed by cones. The cones do not begin to extensively degenerate until almost all rods have been lost [22,23,24]. Quite intriguingly, an age-dependent degeneration of other retinal cell types including cells in the inner nuclear layer (INL) and ganglion cell layer (GCL) has been found in both RP patients and *rd10* mice [22,23,24]. Notably, *rd10* retinal cells of the INL and GCL are not the primary targets of the *Pde6b* gene defect [11,12,13,14,15,16]; the molecular mechanisms of their sequential degeneration are still largely unknown. 

Aberrant ER-Ca^2+^ homeostasis serves as a major player in many other neurodegenerative disorders including Alzheimer’s disease (AD), Huntington’s disease (HD), Parkinson’s disease (PD), and amyotrophic lateral sclerosis (ALS) [25,26,27]. Irrespective of the genetic and anatomical, as well as phenotypic, variability among these diseases, altered ER-Ca^2+^ homeostasis initiates and/or orchestrates pathomechanisms such as proteotoxicity due to misfolded protein aggregation, impaired autophagy, and altered RNA metabolism (reviewed in [25]).

Based on our previous work on these pathomechanisms in age-related neurodegenerative diseases [28,29,30], we examined the time course of the development of hallmarks of abnormal autophagy and of RBPs’ alterations in the photoreceptor and non-photoreceptor cells of the *rd10* retina to determine whether they serve as early pathogenic events. Consistent with this notion, we observed altered ER-Ca^2+^ homeostasis, impaired autophagy, and cytoplasmic RBP aggregation in non-photoreceptor cells of the *rd10* retina by *P17*. At this stage, neurodegenerative changes just start to emerge, without any obvious photoreceptor loss, and these are difficult to detect with light microscopy. The fact that cytoplasmic RBP aggregation occurs in non-photoreceptor retinal neuronal cells before the demise of photoreceptors suggests that functional alterations of connectivity may be sufficient to trigger cytoplasmic RBP mislocalization. These findings in *rd10* mice might benefit our understanding of the spread of pathology in many age-related progressive neurodegenerative diseases that are characterized by Ca^2+^ imbalance, autophagy dysregulation, and altered RBP homeostasis. 

## 2. Materials and Methods

### 2.1. Antibodies

All primary and secondary antibodies and their dilutions used in this study are listed in Appendix A. Many of these antibodies have been used by us in both mice and human tissue in previously published studies (see references in Appendix A).

### 2.2. rd10 Mouse Model

Studies were performed using retinae from male and female *rd10* mice *(C57BL/6J-Pde6b rd10/J)* [11] and age-matched wild-type (WT) mice (C57BL/6) as controls. It is important to note that the two mouse strains share the same genetic background. The breeding of *rd10* mice has been described previously in detail [31]. The animals were kept on a 12 h light/dark cycle with food and water ad libitum. All experiments were performed in accordance with the ARVO statement for the Use of Animals in Ophthalmic and Vision Research and the German Law for the Protection of Animals and after approval had been obtained by the regulatory authorities of the Forschungszentrum Jülich and the Institute of Laboratory Animal Science (Faculty of Medicine, RWTH Aachen University). A4 application number 11280A4.

### 2.3. Preparation of Retinal Sections and Homogenates

Both *rd10* and wild-type mice were deeply anesthetized by isoflurane and sacrificed by decapitation on the following postnatal day: *P16*, *P17*, *P19*, *P22*, *P25*, *P26*, *P50* (*n* = 3 in each age group, otherwise mentioned in the legend). Both the eyeballs were enucleated and used for hematoxylin and eosin staining (H&E), immunohistochemistry, immunofluorescence, transmission electron microscopy (TEM), Western blotting, filter trap assay (FTA), and subcellular fractionation (see also Appendix A). For H&E histology, immunohistochemistry, and immunofluorescence analyses, the eyes were cut open at the limbus. The cornea and lens were removed and the retina in the eyecup was fixed in 4% paraformaldehyde (PFA) for 1 h at room temperature. The unfixed eyeballs of WT mice (*P26*: *n* = 3) and *rd10* (*P17*: *n* = 4, *P26*: *n* = 3) mice were used for Western blotting. For the filter trap assay, a pool of WT (*P26*: *n* = 3) and *rd10* (*P17*: *n* = 3, *P26*: *n* = 3) was used. For the purification of subcellular protein fractions, a Subcellular Protein Fractionation Kit for Tissues (Thermo Scientific, Rockford, IL, USA) was used. Based upon the Western blot analysis performed on these purified protein fractions, the distributions of RBPs in *rd10* and age-matched WT mice were determined. Both the left and right retinae of each of *rd10-P17* (*n* = 3), WT-*P17* (*n* = 3) and WT-*P26* (*n* = 3), *rd10-P17* (*n* = 3), and *rd10-P26* (*n* = 3) were collected (see also Appendix A). For these analyses, the cornea and lens were removed, and the retina was carefully dissected out of the eyecup for homogenization.

### 2.4. Diaminobenzidine (DAB) Immunohistochemistry

Paraffin sections of 3–4 µm were placed on poly-L-lysine coated slides and allowed to dry in an oven at 37 °C overnight and then processed for immunohistochemistry as described in detail elsewhere [30,32]. The sections were deparaffinized in xylene for 20 min and rehydrated in 100%, 96%, and 70% ethanol for 5 min each, followed by endogenous peroxidase quenching (0.3% H_2_O_2_ in methanol) for 20 min. For antigen retrieval, sections were heated in citrate buffer (Dako Target Retrieval solution), pH 6 (Dako, Glostrup, Denmark), for 20 min in a pressure cooker. After washing in PBS, sections were incubated with primary antibody (Appendix A) for 1 h at room temperature or 4 °C overnight. After washing in PBS, sections were incubated with appropriate HRP-linked secondary antibodies (ImmunoLogic, Duiven, The Netherlands) for 30 min at room temperature. DAB reagent (ImmunoLogic, Duiven, The Netherlands) was used to visualize antibody binding. The sections were then counter-stained with 6% Mayer’s hemalum solution (Merck KGaA, Darmstadt, Germany) for 3 min. All procedures were performed at room temperature. Representative sections formed each eye: WT-*P26* (*n* = 3), *rd10-P17* (*n* = 3), and *rd10-P26* (*n* = 3) were analyzed (see also Appendix A).

### 2.5. Immunofluorescence

Single and double immunofluorescence staining were performed as already described [30,32]; representative sections from every genotype were analyzed, (*n* = 3) WT-*P26*, (*n* = 3) *rd10-P17,* and (*n* = 3) *rd10-P26*—and for certain stains, WT-*P50* and *rd10-P50* were analyzed. Briefly, deparaffinized sections were heated in citrate buffer (Dako Target Retrieval solution), pH 6 (Dako, Glostrup, Denmark) for 20 min in a pressure cooker. Sections were then blocked (to avoid non-specific binding) with ready-to-use 10% normal goat serum (Life Technologies, Frederick, MD, USA) for 1 h at room temperature before incubating with primary antibody at 4 °C overnight. After two washes in TBS-T for 10 min, the sections were incubated with Alexa-conjugated secondary antibody (1:500 in TBS-T) at room temperature for 2 h. Sections were then washed in TBS-T (2 × 10 min) and incubated for 10 min in 0.1% Sudan Black/80% ethanol to suppress endogenous autofluorescence. Finally, the sections were washed for 5 min in TBST and mounted with antifade mounting medium with DAPI (Vectashield with DAPI, H-1200, Vector Laboratories Inc., Burlingame, CA, USA).

### 2.6. Image Acquisition and Semi-Quantitative Analysis

Images of the H&E-stained sections (representative sections from each genotype (*n* = 3 WT-*P17*, *n* = 3 WT-*P19*, *n* = 3 WT-*P26*, *n* = 3 WT-*P50*, *n* = 3 *rd10-P17*, *n* = 3 *rd10-P19*, *n* = 3 *rd10-P26,* and *n* = 3 *rd10-P50*) and DAB-stained sections (*n* = 3 WT-*P26*, *n* = 3 *rd10-P17,* and *n* = 3 *rd10-P26)* were taken with a Zeiss Axioplan microscope equipped with a 40× objective and an Axio Cam 506 camera. Images from immunofluorescence labeled sections were taken with a Zeiss LSM 700 laser scanning confocal microscope using 20×, 40×, and 63× objectives. Images were acquired by averaging 4 scans per area of interest resulting in an image size of 1024 × 1024 pixels. The laser intensity was kept constant for all of the samples examined. Captured confocal images were analyzed using Adobe Photoshop CS5 and ZEN (Blue edition) 2009 software. Semi-quantitative analysis (See Appendix A) was performed manually by examining the sections (one representative section) from each of the three mice from each genotype and assigning the following scores based on the pattern of immunoreactivity (+++ strong immunoreactivity/accumulation, ++ moderate immunoreactivity/accumulation, + mild immunoreactivity, ● aggregates, NA, not available/not included in the study).

Semi-quantitative analysis of fluorescence intensity: The average pixel intensity of the target proteins (Calreticulin, SigR1, GRP78, LC3)/field of view were quantified in one representative section from each of three *rd10-P17* and *rd10-P26* and one representative section each from three age-matched wild type controls of WT-*P26*. A total of 8-10 random fields at low magnification (20×) were examined, capturing approx. 15-20-30 ROI with a 40× lens from different parts of the retina (quantification was performed only from the ONL and INL). The average background pixel intensity was subtracted. We used the unpaired Student’s t-test for comparison between two sample groups. Values represent the mean ± standard deviation (SD). Differences between values were regarded as significant when = (* *p* <0.05, ** *p* <0.01).

### 2.7. Western Blot Analysis

Western blots of retina homogenates were performed as described previously [30,32]. Briefly, both the left and right retinae of WT (*P26*: *n* = 3) and *rd10* mice (*P17*: *n* = 4, *P26*: *n* = 3) were homogenized in ice-cold radioimmunoprecipitation assay (RIPA) lysis buffer containing protease inhibitor cocktail (Roche Life Science, Penzberg, Germany). The crude lysates were briefly centrifuged and then processed for the bicinchoninic acid protein assay (BCA; Thermo Scientific, Rockford, IL, USA) according to the manufacturer’s protocol. Equal amounts of protein were boiled in Laemmli sample buffer for 5–10 min and processed for SDS-PAGE. The protein gels were transferred onto polyvinylidene fluoride (PVDF) membranes (Merck KGaA, Darmstadt, Germany). The blots were then blocked with 4% skim milk (Sigma-Aldrich, St. Louis, MO. USA) in TBS-T for 30 min and incubated with a specific primary antibody (Appendix A) overnight at 4 °C under agitation. Thereafter, the blots were washed three times in TBS-T for 10 min each and incubated for 1 h with an appropriate horseradish peroxidase (HRP)-conjugated secondary antibody (Thermo Scientific). Immunoreactive protein bands were visualized by exposing the blots on an X-ray film (Thermo Scientific, Rockford, IL, USA). Quantification of the band intensities was performed after normalizing to tubulin levels using Adobe Photoshop CS5. Values represent the mean ± SD (* *p* < 0.05, ** *p* < 0.01).

### 2.8. Subcellular Fractionation

Briefly, the subcellular protein fractionation kit (Thermo Scientific/Life Technologies) was used to determine subcellular distributions of RNA binding proteins in *rd10* and age-matched WT mice. Both the left and right retinae of each of WT-*P17* (*n* = 3), *rd10-P17* (*n* = 3) WT-*P26* (*n* = 3), *rd10-P17* (*n* = 3), *rd10-P26* (*n* = 3) were collected, pooled, and homogenized (each group containing a total of 6 retinae). Four subcellular fractions were obtained per sample (see below). The standard procedure of subcellular fractionation was performed as previously described [32,33]. In brief, both the left and right retinae were washed gently with ice-cold PBS and processed for subcellular protein fractionation according to the manufacturer’s protocol. The RBP levels were analyzed in the cytoplasmic extract (Ce), membrane extract (Me), soluble nuclear extract (Ne), and chromatin-bound nuclear extract (Cbe) fractions by immunoblotting.

### 2.9. Transmission Electron Microscopy (TEM)

TEM of the retinae of *rd10,* as well as that of age-matched WT, mice were performed using standard protocols [32,33]. Briefly, retinae were fixed in 2.5% glutaraldehyde in 0.1 M phosphate buffer for 24 h followed by washing in the buffer for another 24 h. These samples were then incubated in 1% osmium tetroxide (OsO_4,_ in 0.2 M phosphate buffer) for 3 h, washed twice in distilled water, and dehydrated using ascending alcohol concentrations (i.e., 25%, 35%, 50%, 70%, 85%, 95%, 100%; each step for 5 min). Dehydrated blocks were incubated in propylene oxide followed by a subsequent 20 min incubation in a 1:1 mixture of epon (47.5% glycidether, 26.5% dodenylsuccinic acid anhydride, 24.5% methylnadic anhydride, and 1.5% Tris (dimethyl aminomethyl phenol) and propylene oxide. The samples were then incubated in epoxy resin for 1 h at room temperature followed by polymerization (28 °C for 8 h, 80 °C for 2.5 h, and finally at room temperature for 4 h). Ultra-thin sections (70 nm) were mounted on grids, contrast-enhanced with uranyl acetate and lead citrate, and examined using a Philips CM10 transmission electron microscope equipped with a Morada digital camera as already described [32,33].

### 2.10. Statistics

We used densitometric analysis on the protein bands on the Western blots by using Adobe Photoshop, representing the relative band intensity of the test proteins normalized with tubulin levels. Graphpad prism was used to conduct unpaired Student’s t-tests for comparisons between two sample groups. Values were expressed as mean ± standard error of the mean (SEM) from three independent experiments. The asterisks (*) denote significant differences (* *p* < 0.05, ** *p* < 0.01). The (#) denotes not significant.

## 3. Results

### 3.1. Increased Immunoreactivity and Protein Expression Levels of Calcium-Binding/Buffering Chaperones and ER Stress Markers in the rd10 Retina before any Noticeable Photoreceptor Loss at P17

Photoreceptor degeneration in the *rd10* retina starts at 17–18 days after birth [11,13,14,21]. Consistent with the above studies, there was no obvious difference detected in the overall thickness measurements of the outer nuclear layer (ONL) of *rd10-17* (see white scale bar), and any degenerative changes were hardly noticeable morphologically at this point when compared with the wild-type controls (Figure 1a) by light microscopy. At *P19*, photoreceptor loss became apparent by detectable thinning of the ONL (Figure 1a). At *P26*, the thickness of the ONL was dramatically reduced due to a massive loss of photoreceptors (Figure 1a). At this stage, around 70–80% of the rod and at least 60–70% of the cone photoreceptor cells were already lost [21,22]. At *P50*, nearly all the photoreceptors from ONL had degenerated, while the inner retinal layers (INL and GCL) were barely affected (Figure 1a). In contrast, no neurodegeneration was detected and there was no difference in the staining intensity and pattern of immunoreactivity of any of the test markers (used in this study) among the controls (WT) from *P17*, *P19*, *P26*, and *P50*. 

Considering an overall calcium imbalance as an outcome of the *Pde6b* gene defect, we anticipated the dysregulation of Ca^2+^-associated proteins in photoreceptors at *rd10-P17* and *P26*. In order to confirm this, we stained for the Ca^2+^ binding protein calreticulin, which is involved in Ca^2+^ buffering and storage. We found an increased calreticulin immunoreactivity at *rd10-P17* and *P26* in photoreceptor layers. Interestingly, we also observed increased immunostaining in the somata of the INL and GCL from *rd10-P17* and *P26* compared with WT-*P26* and *P50* (Figure 1b, quantification c), indicating that downstream targets are also affected. In addition, we observed increased calreticulin immunoreactivity in rod bipolar cells of *rd10-P17* and *P26* and even at *P50*, when the photoreceptors were almost completely lost (Figure 1b, lower panel, quantification c). We used PKCα as a marker to detect rod bipolar cells, which are postsynaptic to rods. Their loss was apparent at the later stages of *rd10* neurodegeneration [34,35]. These results were consistent with the Western blot analysis of the calreticulin protein performed on *rd10* retinal lysates (Figure 1i, quantification j). Subsequently, we examined the immunoreactivity patterns of two other Ca^2+^ buffering chaperones, PDI and pPERK. pPERK was also slightly increased in rod bipolar cells and photoreceptor inner segments (below the ONL) (Figure 1d, see also Appendix A for semi-quantitative analysis), while PDI was expressed more strongly in the processes running vertically through the retina, most likely originating from the Müller cells (the major retinal glia cell type) at *rd10-P26* (Appendix A, see also Appendix A for semi-quantitative analysis) in *rd10* mice. Taken together, these results suggest alterations in Ca^2+^ homeostasis already at an early stage before any obvious/major photoreceptor loss. 

Next, we examined whether the ongoing Ca^2+^ imbalance/altered Ca^2+^ homeostasis is associated with increased ER stress levels. For this purpose, we used antibodies against the ER chaperones SigR1, GRP78/Bip, and GADD-153 (Figure 1e,f,h, quantification g). Whilst only a low level of SigR1 immunoreactivity was observed in WT retina, we found significant expression of SigR1 in the photoreceptor inner segments and in INL and GCL somata at *rd10 P17,* and also in photoreceptor somata at *P26* (Figure 1e, quantification g). Immunoreactivity for GRP78/Bip and GADD-153 was only mildly increased (Figure 1f–h, quantification g, see also Appendix A for semi-quantitative analysis). GRP78/Bip was increased in the ganglion cell layer (GCL), suggesting chronic ER stress in these cells. Consistent with the immunohistochemical results, Western blotting analysis revealed significantly increased levels of GRP78/Bip, PDI, and calreticulin in both the *rd10* retinae at *P17* and *P26* compared with the wild-type (WT) at P26, while pPERK and SigR1 protein levels were found to be significantly increased only at *P26* of *rd10* retinae (Figure 1i, quantification j).

### 3.2. Early Autophagy Alterations in the rd10 Retina

Autophagy is known to be tightly linked to the ER and decisive in preventing neurodegeneration mediated by ER stress [36,37]. Therefore, we hypothesized that altered ER-Ca^2+^ homeostasis in the *rd10* retina might lead to alterations in the autophagy process. As expected, immunoreactivities for the autophagy adaptor protein p62 and the autophagosome marker LC3 were significantly increased in the INL and the GCL of *rd10-P17,* and to a lesser extent of *rd10-P26,* compared with WT-*P26* (Figure 2a,b; see also Appendix A for semi-quantitative analysis). In addition, several cells of *rd10-P17* mice harbored cytoplasmic accumulations of p62-positive material/inclusions (Figure 2a, arrows, arrowhead). There was a significantly increased level of LC3 at *rd10-P17*, and a slight increase at *P26*; however, LC3 levels were not increased further at *P50* (Figure 2b). Consistent with the immunolabelling results, immunoblot analysis revealed an overall increase in levels of p62 and increased levels of both LC3I [36] and LC3II in *rd10-P17,* as well as *rd10-P26* retinal lysates (Figure 2d; quantification e; WT-*P26 n* = 3, *rd10-P17 n* = 4, *rd10-P26 n* = 3). Increased levels of both LC3I and lipidated LC3II (not shown) and altered levels of the ratio between LC3II/LC3I were suggestive of altered autophagy, most likely at both the initial (autophagosome formation/maturation) and final steps (autophagosome fusion to lysosomes) [38,39,40]. Neurodegeneration along with marked autophagy impairment in photoreceptors was later confirmed by the presence of large abnormal cytoplasmic autophagic vacuoles filled with membranous and granular debris in photoreceptor inner segments (IS) and somata in the ONL of *rd10* mice at *P19* (Figure 2f) and at *P22* (Figure 2g) examined by electron microscopy. Such vacuoles were absent from WT controls (not shown). Taken together, these data show that signs of ER stress and defective autophagy are already manifested in photoreceptor and non-photoreceptor cell populations in the *rd10* retina at early stages of photoreceptor degeneration.

### 3.3. Abnormal Cytoplasmic Aggregation of pTDP-43 and Matrin 3 at Early Stages of rd10 Retina Degeneration

ER-Ca^2+^ dynamics can efficiently regulate autophagy as well as RBP homeostasis; on the other hand, aberrant ER-Ca^2+^ may serve as a major driver for RBPs’ alterations including TDP-43-mediated neuronal toxicity [41,42]. Similarly, the effective turnover/clearance of many disease-associated RBPs such as TDP-43 and FUS is regulated by the ubiquitin-proteasome system (UPS) and autophagy [43,44]. Thus, we aimed to determine whether ongoing ER-Ca^2+^ autophagy defects are associated with altered distribution of key RBPs linked to neurodegenerative disease, in particular TDP-43, FUS, and Matrin 3. These RBPs normally reside in the nucleus. In pathological/neurodegenerative conditions such as amyotrophic lateral sclerosis/frontotemporal lobar degeneration (ALS/FTLD), they often accumulate in the cytoplasm of affected neurons [45,46,47]. Using DAB immunohistochemistry, we found globular cytoplasmic pTDP-43 aggregates already present at *P17* throughout the retinal cell layers including many photoreceptors, INL cells, and ganglion cells (Figure 3a; see also Appendix A for semi-quantitative analysis). pTDP-43 aggregates were also evident at considerable numbers at *P26*, but there was no further increment regarding their number nor their staining intensity (Figure 3a, right panel, see also Appendix A for semi-quantitative analysis) compared with *P17*. Immunofluorescence using the same antibody showed a higher sensitivity and revealed the accumulation of pTDP-43 in more than one-third of the photoreceptors (Figure 3b) at *P17* and nearly 40% at *P19* in the ONL, when the acute degeneration of the photoreceptors was the most pronounced (not shown). In line with the results obtained from the sections stained with DAB-immunohistochemistry mentioned above, pTDP-43 aggregation did not increase further at *P26* nor at *P50* (not shown) compared with *P17*. Interestingly, both DAB and immunofluorescence labelling consistently depicted pTDP-43 immunoreactive inclusions also present in cells at the outer margin of the inner nuclear layer. We recognized these cells as horizontal cells by using co-immunolabelling against the CabP antibody (Figure 3c).

The expression of TDP-43 is tightly regulated by another closely related RBP, Matrin 3 [48]. We found a mild cytoplasmic and stronger nuclear accumulation of Matrin 3 in *rd10* mouse ONL, INL, and GCL cells at *rd10-P17* and *P26* (Figure 3d, arrows). Again, immunofluorescence labeling using the same antibody was very sensitive and revealed an altered pattern of Matrin 3 immunoreactivity, with increased granular nuclear accumulation of Matrin 3 in many cells (Figure 3e, arrows, right panel); on the other hand, several adjacent cells showed reduced levels of nuclear Matrin 3 immunoreactivity (Figure 3e, red arrowheads). Interestingly the photoreceptors showed an overall increase in the nuclear accumulation of Matrin 3 (arrows) in *rd10-P17* compared with *WT-P17* (Figure 3f, arrows). These results were further verified by Western blot analysis of subcellular fractions obtained from the retinal lysates (Figure 3g, red arrowheads; a pool of *n* = 3 WT-*P17* and *n* = 3 *rd10-P17*, and Figure 3h, red arrowhead, pool of *n* = 3 WT-*P26*, *n* = 3 *rd10-P17,* and *n* = 3 *rd10-P26*), see Section 2 (Methods for details).

### 3.4. Aggregation of FET Family Proteins (FUS, EWRS1, and TAF15) in rd10 Retina

Members of the FET protein family (FUS, EWSR1, and TAF15) bind to RNA/DNA and regulate transcription, RNA processing, and other aspects of RNA/DNA homeostasis [49,50,51,52]. FET proteins are involved in several neurodegenerative diseases as they can also form toxic aggregates, and perturb protein homeostasis, thus driving neurodegeneration [49,52]. We observed globular cytoplasmic FUS protein accumulation in both photoreceptors and many cells of the INL (Figure 4a, white arrows). Focal intra-nuclear FUS-positive accumulations, as well as strongly labeled FUS-positive condensed/pyknotic nuclei (Figure 4a, black arrows), were also evident. Similar to the pattern of FUS immunoreactivity, other FET family member proteins such as hnRNPA1, hnRNPA2B1, and EWSR1 often showed focal intranuclear accumulation (Figure 4b–d, black arrows). hnRNPA1 and EWSR1, however, did not show significant cytoplasmic accumulation at *P17*, but at *P26,* several cytoplasmic globular aggregates were visible in photoreceptors and INL cells (Figure 4b–d, white arrows). Here again, immunofluorescence showed a higher sensitivity; large globular FUS immunoreactive aggregates were extensively observed in the photoreceptors of *rd10-P17* mice (Figure 4e, white arrows, upper left). Similarly, both focal intranuclear accumulation (Figure 4e, orange arrows, upper right), as well as cytoplasmic inclusions of FUS in rod bipolar cells (labeled for PKCα) (Figure 4e, white arrows), were evident in *rd10* mice at *P17*, *P19,* and *P26*. In addition to this, large FUS-immunoreactive aggregates were also evident in GCL cells at *rd10-P17* (not shown) and *P26* (Figure 4e, white arrow, lower panel). The formation of SDS-resistant FUS aggregates in the *rd10* retina was further confirmed by using a standard filter trap assay (Figure 4f). Consistent with this, subcellular fraction analysis using Western blots of retinal lysates further confirmed the cytoplasmic mislocalization (red arrowhead) of FUS, hnRNPA1, and TAF15 in *rd10-P17* and *rd10-P26* retinae (pool of *n* = 3 WT-*P26*, *n* = 3 *rd10-P17,* and *n* = 3 *rd10-P26*, Figure 4g).

### 3.5. Formation of RNA Stress Granules (SGs) in rd10 Retina

The aggregation of the above-mentioned RBPs mostly proceeds through the SG pathway and might thus serve as a seed for further pathological aggregates, if persistent [46,47,53,54,55]. Therefore, we asked whether the accumulations of the RBPs studied were accompanied by SG formation. We found small globular cytoplasmic aggregations of the widely studied SG protein Tia1 in photoreceptors and INL cells of *rd10-P17* and *P26* mice (Figure 5a, white arrows), together with focal intranuclear accumulation (Figure 5a, black arrows). Interestingly, Tia1-positive SG accumulations were abundant especially at early stages of retinal degeneration (*P17*), and they appeared less frequently at a later stage (*P26*) (Figure 5a, right panel *P26*). Co-immunolabelling using a combination of Tia1 and FUS or TDP-43 antibodies revealed a clear pattern of sequestration of Tia1 with FUS and TDP-43 within these aggregates (Figure 5b, arrows). Consistent with this, subcellular fraction analysis using Western blots of retinal lysates further confirmed the cytoplasmic accumulation of Tia1 (red arrowheads) at *P17* and *P26* (pool of *n* = 3 each of WT-*P26*, *rd10-P17,* and *rd10-P26*, Figure 5c). These results further confirm the notion that RBP aggregates proceed through the SG pathway and that aberrant SGs containing RBPs can be considered to be an early pathological event in the aggregation of RBPs.

## 4. Discussion

In the commonly used *rd10* mouse model of retinitis pigmentosa, a mutation in the Pde6 gene leads to photoreceptor degeneration and consequent blindness [11,12,13,14,15]. Milder and slower degeneration in certain second and third-order neurons also occurs. As Pde6 is exclusively expressed in rod photoreceptors, this degeneration must be a secondary effect, e.g., as a consequence of the lack of synaptic input by the photoreceptors [11,12,13,14,15,21]. However, the pathomechanisms of the degeneration of non-photoreceptor neuronal cell types in RP are poorly understood. Photoreceptor degeneration in the *rd10* retina starts at 17–18 days after birth [11,12,13,14,15,21]. At *P17*, all cell layers still appear anatomically unchanged compared with age-matched WT retina (Figure 1), and light responses can be recorded on the level of the electroretinogram or individual ganglion cells [14,56]. Interestingly, at *P17*, we could already find the expression of several markers commonly observed in other types of neuronal degeneration, not only in photoreceptors but also in cells of the INL and GCL. We found (a) increases in the Ca^2+^ buffering chaperones calreticulin, PDI, and pPERK, together with the ER chaperones SigR1 and GRP78/Bip; (b) altered expression of the autophagy proteins p62 and LC3, together with the accumulation of autophagic vacuoles; (c) cytoplasmic and nuclear aggregation of RBPs such as pTDP-43 and FET-family RBPs (see also schematic representation Figure 6). In the early stage, only rod photoreceptors die, followed by cones later in the process. In line with this, we found that the two cell types that receive direct input from rods and which display considerable loss at later stages of retinal degeneration [34,35], i.e., the rod bipolar cells and the horizontal cells (see, e.g., Figure 1 and Figure 3), express the markers summarized above. These results suggest that deleterious effects due to altered Ca^2+^ homeostasis and ER stress initiated in rods lead to similar alterations in interconnected cell types. From this point of view, it appears logical that these cell populations upregulate levels of neuroprotective ER chaperones. In fact, SigR1 is extensively studied for its neuroprotective role in retinal cells [57,58,59,60].

Above-threshold Ca^2+^ imbalance and ER stress that cannot be managed by the above-mentioned factors should eventually lead to the failure of cellular proteostasis networks including protein quality control/autophagy pathways, resulting in misfolded protein aggregation and associated neurotoxicity (see also schematic representation Figure 6). This has been observed in major chronic neurodegenerative disorders including Alzheimer’s disease (AD), Huntington’s disease (HD), Parkinson’s disease (PD), and amyotrophic lateral sclerosis (ALS) [25,61,62]. Altered proteostasis has also been previously discussed as a factor contributing to the spread of degenerative pathology in RP retina cell populations [15,21]. The results of the present study add credibility to this concept. They add the surprising observation that abnormal protein aggregates can already be observed at the early stages of neurodegeneration, not only in photoreceptors but also in non-photoreceptor cell types, and that they include aggregates of RBPs—most prominently of TDP-43, an important player in the pathophysiology of several major neurodegenerative diseases including ALS/FTLD [45,52]. Additionally, toxic modifications of these protective proteins caused by chronic cellular stress conditions could contribute to progressive neurodegenerative phenotypes as seen in *rd10* retina [15,21,22,63].

RBPs regulate several crucial aspects of neuronal gene expression including splicing regulation, mRNA transport, and modulation of mRNA translation and decay. In addition, RBPs are known to regulate proteostasis and autophagy at multiple steps [45]; in turn, the efficient turnover of several disease-associated RBPs including TDP-43, FUS, and SGs is regulated by UPS and autophagy [43,44]. One of the intriguing aspects observed here is that the build-up of RBP aggregates is happening rather early and quickly, along with the build-up of altered ER-Ca^2+^ homeostasis and ER stress. Thus, it appears reasonable to assume that the ongoing chronic cellular stress conditions may quickly lead to defects in the nucleus regarding cytoplasmic shuttling and thus to impaired RBP homeostasis [45,46,47,50,53].

Altered RBP homeostasis and defective proteostasis/autophagy are key mechanisms triggering neurodegeneration in many neurodegenerative diseases including ALS [45,46,47,50,53]. We observed both altered autophagy and RBP aggregations at early stages throughout the retinal cell layers, even though the *Pde6b* gene defect selectively affects rod photoreceptor cells. Moreover, the *Pde6b* gene does not primarily target RBP metabolism. The RBPs studied here, including TDP-43 and FET proteins such as FUS and EWSR1, are particularly susceptible to aggregation due to the presence of both RNA-binding domains and prion-like domains enriched in uncharged polar amino acids (such as asparagine, glutamine, and tyrosine), which contribute strongly to aggregate formation under stressful stimuli including ER-Ca^2+^ imbalance, ER-stress, and autophagy impairment [50,51,55,64]. These findings are also reminiscent of various age-related neurodegenerative diseases such as AD, PD, HD, and prion diseases, where specific subsets of neurons are affected and the disease-causing protein aggregates initially appear in a restricted distribution and only later spread throughout the CNS [42]. We observed a close association of RBP build-up with stress granule formation in affected neuronal cells, especially at an early stage (P17) of retinal degeneration in rd10 mice, suggesting that these aggregates arise de novo in these cells.

Further studies will be required to determine how the *Pde6b* gene mutation leads to this cascade of events. Regardless of the initial events, considering the hierarchical cluster of heterogeneous types of neurons and interconnected, interdependent units, we cannot exclude the possibility that a prion-like seeding and spreading of the above-mentioned RBPs could be implicated further downstream, similar to the spreading of RBP aggregates in other neurodegenerative diseases [65,66].

We conclude that—owing to the clear separation of cell types affected by primary and secondary degeneration—the retina provides an exquisite model to study neuronal degeneration processes, in particular as many similarities were found to processes described for other neurodegenerative disorders. Progressive neurodegeneration in the *rd10* mouse retina is associated with early disturbances of proteostasis and autophagy and with abnormal cytoplasmic RBP aggregation and stress granule formation. These processes are already triggered in the very early phases of retinal degeneration (see also schematic representation Figure 6).

## Figures and Tables

**Figure 1 cells-12-01094-f001:**
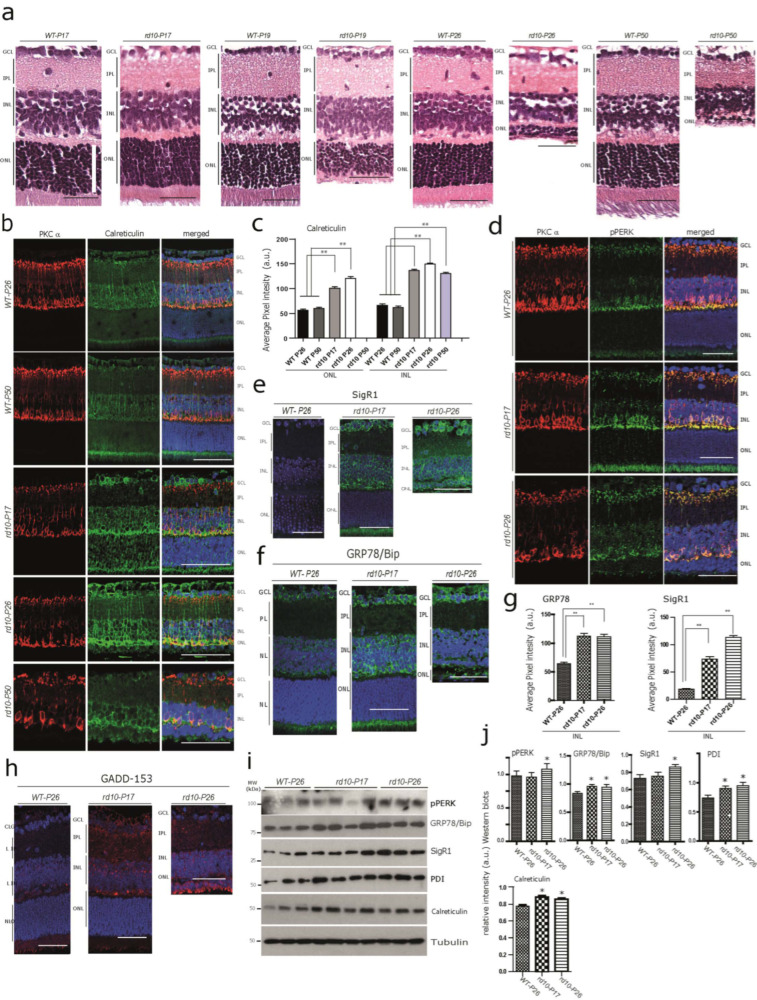
Early alterations of Calcium-Binding/ ER Chaperones in the rd10 Retina (**a**) Hematoxylin–eosin (H&E)-stained retinal paraffin sections showing age-dependent progressive neurodegeneration in *rd10* mice at *P17, P19, P2,6* and *P50* compared with age-matched wild-type mice. The retinal cell layers: the outer nuclear layer (ONL), inner nuclear layer (INL), inner plexiform layer (IPL), and ganglion cell layer (GCL). Note the lack of noticeable degeneration in the cells of ONL, INL and GCL in *rd10-P17* compared with the WT-*P17* (white scale bar—70 µm). In the WT retina (WT-*P17*, WT-*P19*, WT-*P26*, and WT-*P50*), there were no differences in the staining pattern of any markers used in this study. Thus, throughout the remainder of the manuscript, only one representative WT-control retina is shown for comparison. Scale bars = 50 µm. (**b**,**c**) Double immunofluorescence using the rod bipolar cell marker, protein kinase c alpha (PKC α) with calreticulin ((**b**), quantification (**c**)), and with pPERK (**d**). Note the increased immunoreactivity for calreticulin and the mild-to-moderate increase in pPERK immunoreactivity observed in many retinal cell layers in *rd10* mice, especially at *rd10-P17*, but also at *rd10-P26* compared with WT-*P26* and WT-*P50*. (**e**–**h**) Increased levels of the ER chaperones sigma receptor 1 (SigR1) (**e**) and GRP78/Bip (**f**), and activation of the ER stress marker GADD-153 (**h**) in the *rd10-P17* mouse retina. (**b**,**d**–**h**) Images from a single representative section from *n* = 3 WT-*P26*, *n* = 3 WT-*P50*, *n* = 3 *rd10-P17*, *n* = 3 *rd10-P26,* and *n* = 3 *rd10-P50*. Scale bars (**d**–**h**) = 45 µm. (**c**,**g**) Quantifications for **(b**,**e**,**f**). (**i**) Immunoblot analysis of retinal tissue homogenates from *rd10* and WT mice of the calcium-binding/buffering chaperones pPERK, PDI, and calreticulin and of the ER chaperones SigR1 and GRP78/Bip in early (P17) and peak (P26) stages of *rd10* retina degeneration compared with WT-P26. (**j**) Quantitative densitometry analysis of relative band intensities (WT-P26 *n* = 3, *rd10*-P17 *n* = 4, *rd10-P26 n* = 3). The asterisks (*) denote significant differences (* *p* < 0.05, ** *p* < 0.01).

**Figure 2 cells-12-01094-f002:**
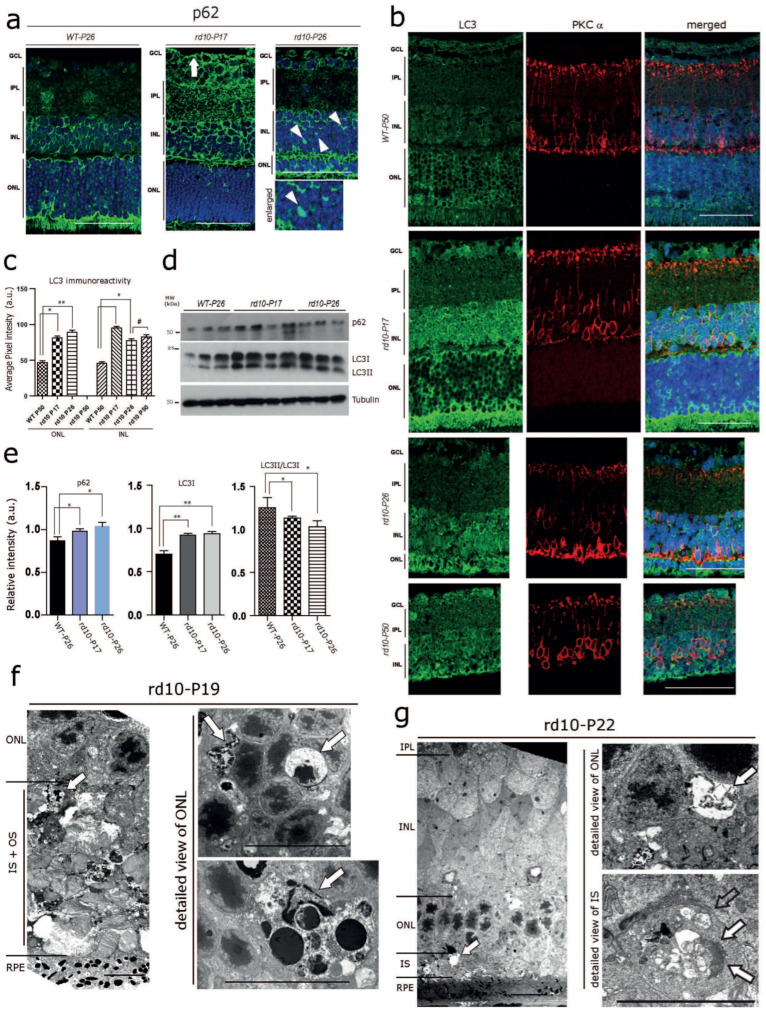
Early Autophagy Alterations in the rd10 Retina (**a**) Immunofluorescence labelling of the autophagy adaptor protein p62. Markedly increased p62 levels in the INL, the inner plexiform layer (IPL), and the GCL (white arrow) at *rd10-P17,* and a moderate increase at *rd10-P26* compared with WT-*P26*. Note the p62-positive globular cytoplasmic accumulations in the INL in the *rd10-P26* retina (white arrowheads). (**b**) Increased LC3 immunoreactivity in PKCα-positive rod bipolar cells at *rd10-P17* and *P26* and in the ONL of *rd10-P26* mice. Images from a single representative section taken from *n* = 3 *WT-P50*, *n* = 3 *rd10-P17*, *n* = 3 *rd10-P26,* and *n* = 3 *rd10-P50*. (**c**) Quantification for (**b**). Scale bars (**a**,**b**) = 40 µm. (**d**) Immunoblot analysis of retinal tissue homogenates showing slightly increased overall levels of p62 and markedly increased levels of both LC3I and LC3II in *rd10-P17* and *rd10-P26* retinas compared with WT-*P26* (WT-*P26 n* = 3, *rd10-P17 n* = 4, *rd10-P26 n* = 3). (**e**) Quantification of p62 (below) and LC3 immunoblots (middle and right). Note the decrease in the ratio between the lipidated form of LC3II/LC3I (right). (**f**,**g**) Electron micrographs showing neurodegeneration and cytoplasmic accumulations of vacuolar material (arrows) at *rd10*-*P19* (**f**) and at *rd10*-*P22* (**g**). Note the large, abnormal autophagic vacuoles filled with membranous and granular osmiophilic material (arrows) in the somata of the ONL and inner segments (IS) of photoreceptor cells. Scale bars= left panel: 10 µm, detailed view in the right panel: 2 µm. The asterisks (*) denote significant differences (* *p* < 0.05, ** *p* < 0.01). The (#) denotes not significant.

**Figure 3 cells-12-01094-f003:**
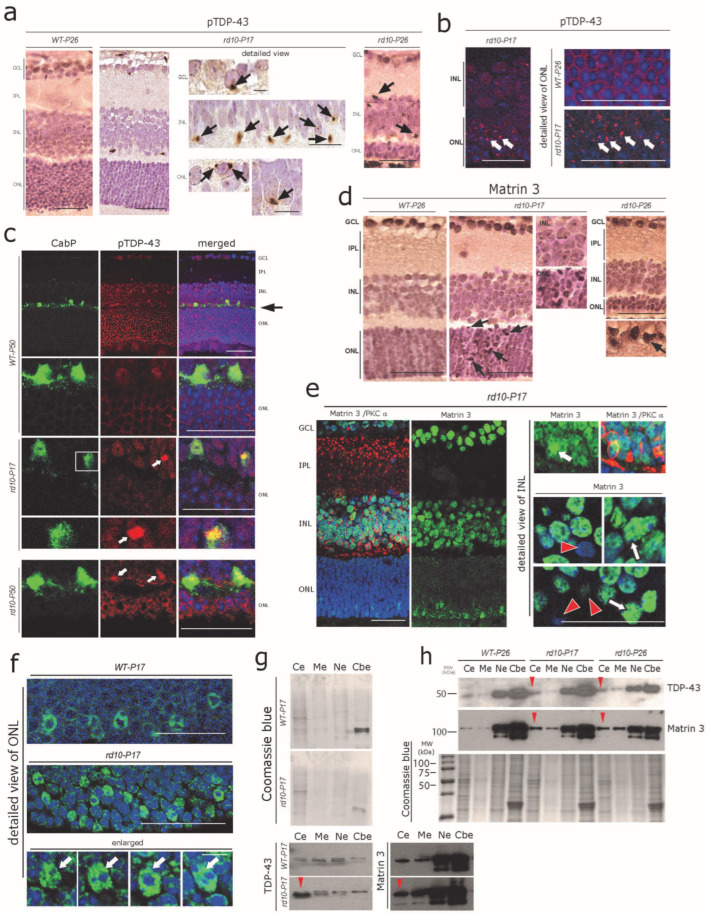
Early cytoplasmic aggregation of pTDP-43 and Matrin 3 in rd10 retina (**a**) Small globular pTDP-43-immunoreactive cytoplasmic aggregates in ONL, INL, and GCL cells of *rd10-P17* and *rd10-P26* mice (arrows). Such pTDP-43-immunoreactive aggregates were absent in WT-*P26* mouse retinas. (**b**) Immunofluorescence staining showing massive aggregation (arrows) of pTDP-43 in the cells of the ONL of *rd10-P17* (right panel: detailed view), while such pTDP-43 aggregates are absent in WT-*P26* control (right upper, enlarged). (**c**) CabP-positive horizontal cells (green) harbor pTDP43-immunoreactive cytoplasmic aggregates (white arrows) in *rd10-P17* (inset enlarged) and *rd10-P50* mice. Scale bars = 40 µm (**d**) Mild-to-moderate nuclear accumulation of Matrin 3 in cells of the ONL, INL, and GCL of *rd10-P17* and *P26* mice (arrows in ONL) as shown by DAB immunohistochemistry. Scale bars = 50 µm (**e**) General overview of altered Matrin 3 immunofluorescence in the *rd10*-*P17* retina (left panel). Note the loss of nuclear localization in some cells (red arrows, lower right) and strong nuclear granular accumulation (white arrows) in other cells of the INL of *rd10-P17* retinae. Scale bars = 40 µm (**f**) Photoreceptors in the ONL show an overall increase in the nuclear accumulation of Matrin 3 (arrows) in *rd10-P17* compared with *WT-P17* mice. (**a**–**f**) Representative sections from *n* = 3 WT-*P26*, *n* = 3 WT-*P50*, *n* = 3 *rd10-P17*, *n* = 3 *rd10-P26,* and *n* = 3 *rd10-P50* are depicted. Detailed view panel (**e**,**f**) = 15 µm. (**g**,**h**) Subcellular fraction analysis of retinal lysates, pool of *n* = 3 WT-*P17*, *n* = 3 *rd10-P17* (**g**), and *n* = 3 WT-*P26*, *n* = 3 *rd10-P17,* and *n* = 3 *rd10-P26* (**h**). Note the cytoplasmic mislocalization of TDP-43 and Matrin 3 in *rd10-P17* (**g**) and in both *rd10-P17* and *rd10-P26* (**h**) (red arrowheads). Cytoplasmic extract (Ce), membrane extract (Me), soluble nuclear extract (Ne), and chromatin-bound nuclear extract (Cbe) fractions by immunoblotting.

**Figure 4 cells-12-01094-f004:**
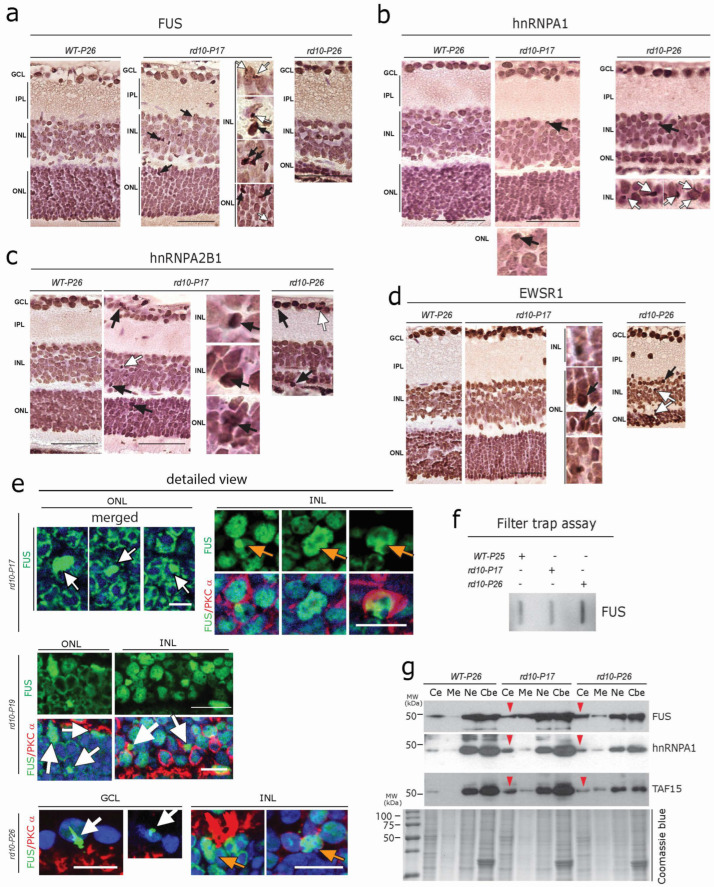
Early aggregation of FET Family Proteins in rd10 retina (**a**–**d**) DAB immunolabelling using antibodies against the FET family proteins FUS (**a**)**,** hnRNPA1 (**b**), hnRNPA2B1 (**c**), and EWSR1 (**d**) revealed globular cytoplasmic accumulation (white arrows in (**a**–**d**)) in photoreceptors of the ONL and many cells of the INL. Focal intranuclear accumulations (black arrows), as well as condensed/pyknotic nuclei, were also evident. A single representative section from every genotype (*n* = 3 WT-*P26*, *n* = 3 WT-*P50*, *n* = 3 *rd10-P17*, *n* = 3 *rd10-P26,* and *n* = 3 *rd10-P50*) is depicted. (**e**) Immunofluorescence revealed large globular FUS-immunoreactive aggregates in the photoreceptors of *rd10-P17* mice (white arrows) and focal intranuclear accumulation (orange arrows), as well as cytoplasmic inclusions of FUS (white arrows, lower panels) in rod bipolar cells of the INL at *rd10-P17* and *P19*. Note the large FUS-immunoreactive aggregates in GCL cells at later stages (*rd10-P26*) (white arrow, lower panel) together with the focal intranuclear accumulation in INL cells (orange arrows). Scale bars = (**a**–**d**) 60 µm, (**e**) 15 µm (**f**) Filter trap assay showing SDS-resistant FUS aggregates in *rd10-P26*. (**g**) Subcellular fraction analysis of retinal lysates, showing the cytoplasmic mislocalization of FUS, hnRNPA1, and TAF15 in *rd10-P17* and *rd10-P26* (red arrowheads): a pool of *n* = 3 WT-*P26*, *n* = 3 *rd10-P1,7* and *n* = 3 *rd10-P26*. Cytoplasmic extract (Ce), membrane extract (Me), soluble nuclear extract (Ne), and chromatin-bound nuclear extract (Cbe) fractions by immunoblotting.

**Figure 5 cells-12-01094-f005:**
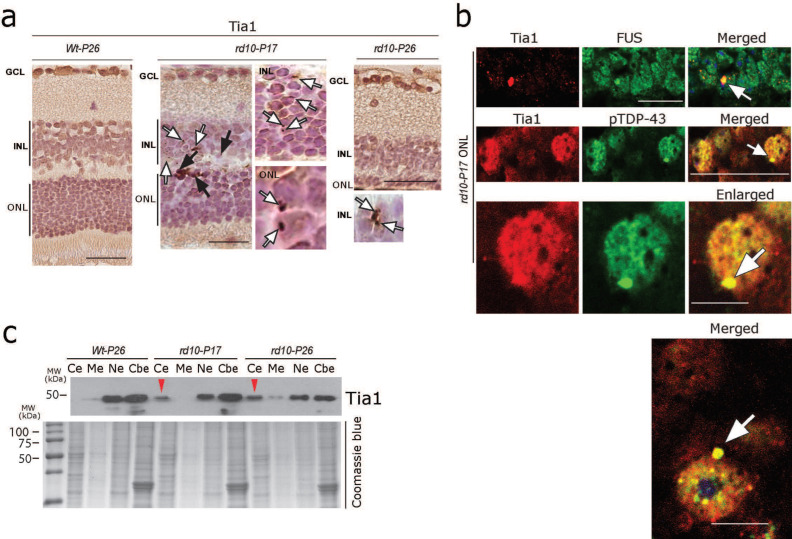
Formation of RNA stress granules in *rd10* retina. (**a**) DAB immunohistochemistry showing a large globular accumulation (black arrow) and small granular aggregation of Tia1 (white arrows) in the ONL and INL cells of *P17* and *P26 rd10* mice. Single representative sections are depicted from each genotype; *n* = 3 WT-*P26*, *n* = 3 *rd10-P17*, *n* = 3 *rd10-P26*, *rd10-P50*. Scale bars = 30 µm. (**b**) Co-immunolabelling showing the colocalization of FUS and pTDP-43 with SG marker Tia1. Scale bars = 20 µm, scale bars = 5 µm enlarged figure. (**c**) Subcellular fraction analysis of retinal lysates showing the cytoplasmic mislocalization of the stress granule protein/marker Tia1 in a pool of *n* = 3 WT-*P26*, *n* = 3 *rd10-P17,* and *n* = 3 *rd10-P26.* Cytoplasmic extract (Ce), membrane extract (Me), soluble nuclear extract (Ne), and chromatin-bound nuclear extract (Cbe) fractions by immunoblotting.

**Figure 6 cells-12-01094-f006:**
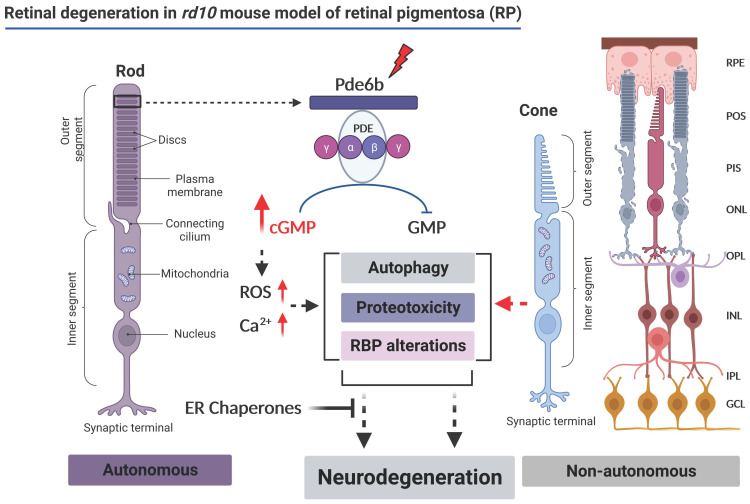
Schematic representation of the proposed molecular mechanism associated with the retinal degeneration in a *rd10* mouse model of retinitis pigmentosa (modified from [11,12,13,14,15,16], image created in BioRender (by BioRender.com).

## Data Availability

The data presented in this study are available on request from the corresponding author.

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
