# Peer review of "Early Alterations of RNA Binding Protein (RBP) Homeostasis and ER Stress-Mediated Autophagy Contributes to Progressive Retinal Degeneration in the rd10 Mouse Model of Retinitis Pigmentosa (RP)"

_cells, 2023, doi:10.3390/cells12071094_

Round 1
Reviewer 1 Report
Please, find the replies attached.

Author Response
March 6th, 2023
Dear Editors,
We are grateful for the constructive review of our manuscript entitled “Early alterations of RNA binding protein (RBP) homeostasis and ER stress-mediated autophagy triggers progressive retinal degeneration in the rd10 mouse model of retinitis pigmentosa (RP)”. please find below our point-by-point responses to the reviewers´ comments. The new version of our manuscript has been revised according to the reviewers´ suggestions (changes highlighted). We sincerely hope that you will find the revised version acceptable for publication.
Yours sincerely,,
Anand Goswami and Joachim Weis
Comments and Suggestions for Authors
Reviewer 1
Comment
The authors describe using the rd10 mouse model of RP to investigate the association of RNA binding protein (RBP) aggregation and abnormal autophagy with early pathogenic events in the retina. The authors examine levels of various markers of autophagy, RBPs, FET proteins, and RNA stress granules using retinal samples. The authors show these markers are increased before the onset of photoreceptor degeneration in the rd10 mouse model. Thus, the authors conclude that there is an association between RBP aggregation and abnormal autophagy with photoreceptor degeneration in the rd10 mouse.
Although the data presented in this manuscript supports the authors’ claims, a major issue with this work is the lack of establishing a causative relationship between RBP aggregation and abnormal autophagy with the photoreceptor degeneration in the rd10 mouse.
Response
We thank the reviewer for presenting his/her general overview of the manuscript and highlighting our major findings. We raised the hypothesis that progressive neurodegeneration in the rd10 retina is mechanistically linked to pathogenic alterations of key RNA binding proteins (RBPs) in the context of defective autophagy. Indeed, we found that progressive neurodegeneration in the rd10 mouse retina is associated with early disturbances of proteostasis and autophagy and with abnormal cytoplasmic RBP aggregation in addition to overall altered ER-Ca2+ homeostasis. Notably, cytoplasmic mislocalization and aggregation of the RBPs; TDP-43, FUS, and Matrin 3 was an early feature of the spread of pathology to non-photoreceptor cell types and was linked to stress granule formation.
We agree with the reviewer that we did not perform further functional experiments to establish a causative relationship between RBP aggregation and abnormal autophagy with the photoreceptor degeneration. Such experiments would have been beyond the scope of the current study, and would certainly not be feasible within time limits of a manuscript revision process. On the other hand, we believe that our manuscript provides a considerable dataset and meaningful new aspects. Remarkably, cytoplasmic aggregation of RBPs including pTDP-43, FUS and Matrin-3 linked to stress granule formation is a characteristic pathognomonic feature of various other neurodegenerative diseases including ALS, AD, and PD. It has been studied in great detail in the context of these disorders and has been found to be central to the neurodegenerative process. Thus, our novel findings are suggestive of a major mechanism of neurodegeneration in the rd10 mouse retina which links this mouse model to other neurodegenerative disorders.
Comment
The authors rely on the lack of photoreceptor degeneration at P17 in the rd10 mouse to make their conclusions but there is a report that has found significant ERG deficits in these mice at P14, even though there is no photoreceptor degeneration at this age (PMID: 24683495). Thus, it is unknown whether the RBP aggregation and abnormal autophagy induce the photoreceptor degeneration in the rd10 mouse or occur as a result of other pathogenic mechanisms that could explain the retinal phenotypes of these mice.
Response
Neuropathological assessments of progressive neurodegeneration including the electroretinography on the rd10 retina are well documented in many studies [1-5] and they are cited in our manuscript. Thus, based on their standard neuropathological assessments of age-dependent progressive neurodegeneration and consistent with our own microscopic examinations of multiple/serial HE stained sections at P17, there is minor, if any neurodegeneration (termed unnoticeable by light microscopy in our manuscript).The degenerative process becomes evident by P19 in terms of retinal thickness, width of the outer nuclear layers, and/or nuclei per row counts. Thus, here in this study we have pinpointed the early alterations of several ER-chaperones/proteins, aggregations of RBPs at a very early stage without obvious/major photoreceptor loss (P17 here in this case).
Interestingly, several members of our group (SJ, FM, PW) had been involved in the study mentioned (PMID 24683495, Rösch et al., 2014) [6], where they showed that the light response in the ERG of rd10 retina (reduction of the a- and b-waves) is diminished already before P17. Based on these data the reviewer might have gained the impression that the neurodegeneration occurs earlier than P17 and differences in protein expression and aggregation that we have described might not be the cause but the result of cellular degeneration before P17. However, the main observation reported in the paper by Rösch et al. [6] is the reduction of the a and b-waves in the ERG of rd10 mice already at earlier stages. In rd10 retina, a point mutation in the Pde6b gene dramatically decreases PDE6 activity, caused by a decrease in protein stability and reduced transport of PDE6 protein to the outer segment. PDE6 is a crucial element in the photo-transduction cascade. It is conceivable that a reduction of PDE6 activity reduces the efficiency of the photo-transduction cascade, leading to smaller light responses. We think that the reduced ERG responses observed by Rösch et al. indicate reduced efficiency of phototransduction rather than photoreceptor degeneration occurring before P17.
Other Minor Comments:
Introduction:
Lines 53-55: Cite references that support this sentence.
Response: Corrected
Lines 55-56: Cite references that support this sentence.
Response: Corrected
Lines 56-57: Cite references that support this sentence.
Response: Corrected
Line 73: Remove hyphen in dis-ease
Response: Corrected
The authors could measure the thickness of the ONL in Figure 1A to support conclusions.
Response: As mentioned above we do not meant to confirm or re-establish the already well-established stages of neurodegeneration in rd10 mice [1-5], but rather to pinpoint the early alterations of several ER-chaperones/proteins, aggregations of RBPs at a very early stage (P17 here in this case), where the neurodegenerations have just started but are yet difficult to detect with light microscopy. We used Axio-vision software and their plug ins for the measurements to scale the ONL and there was no difference between the measured thickness of ONL between Wt-P26 and rd10-P17. We have now included this information in Figure 1a.
The authors should include data on WT-P50 and rd10-P50 in Figure 1G since these age points are mentioned in the results.
Response: Included
The authors could combine the representative images and quantification of the images closer to each other in Figure 1 (for example, combine panels B and G, panels D and G, and panels E and H).
Response: Corrected as suggested
Did the authors verify the increased calreticulin staining with increased calreticulin protein on a Western blot?
Response: We thank the reviewer for this suggestion. We performed and included the WB analysis for calreticulin and consistent with the increased immunoreactivity of calreticulin in the IHC shown in Figure 1b we also observed increased levels of this protein in the WB analysis of rd10 retinal lysates.
The authors observed changes in SigR1 staining at both P17 and P26 ages but only in total SigR1 protein through Western blot analysis at P26. Can the authors explain these differences? Maybe this should be included in this section of the results?
Response: We thank the reviewer for pointing out this specific observation which we failed to adequately mention in the previous version of the manuscript. The SigR1 antibody used for IHC is very sensitive and is capable of detecting minor/moderate changes and has been used by us in several previous studies (References in Table S1). On the other hand, even though we observed a pattern of increased levels of SigR1 detected by the other antibody used for WB analysis, this antibody can only detect major changes. We have now mentioned this in the Results section.
Lines 269-272: The authors only found differences in GRP78 and PDI at P17 (not P16 as described in this paragraph) and not in PDI and pPERK. The authors need to revise this sentence accordingly.
Response: Corrected
Figure 1: The magnification of images is missing and should be added.
Response: Included
The authors do not include the quantification of their P62 western blot in Figure 2.
Response: Included
The authors should analyze the LC3I and LC3II bands separately and compare the amount of lapidated LC3II to LC3I since this ratio is more telling about the status of autophagy in the retinas of the mice than the overall protein amounts of LC3.
Response: We thank the reviewer for the valuable suggestions. We now included a new WB analysis as well as new quantifications. Based on these results, we propose altered autophagy at both initial (autophagosome formation / maturation) and final steps (autophagosome fusion to lysosomes).
What are the magnifications of the images presented in Figure 2?
Response: We regret the confusion, now the magnifications are highlighted in yellow
Can the authors provide and a quantitative analysis of the images presented in panel b of Figure 2? This would strengthen the authors’ conclusions.
Response: Included
Can the authors provide a quantitative analysis of the images presented in Figure 3? This would strengthen the authors’ conclusions.
Response: We thank the reviewer for this valuable suggestion. Figure 3 deals with the aggregation and cytoplasmic mislocalization of RBPs including pTDP-43 and Matrin 3. The semi-quantitative analysis for the immunoreactivity pattern is provided in Table S2.
Can authors clarify if the sample size in Figure 3h refers to number of retinas used in generating fractions or number of independent samples used for western blotting?
Response: Although we have mentioned the sample size in the methods as well as in the Figure legend sections, we agree that this was not adequately elaborated. This aspect is now highlighted both in the Methods and in the Figure legend sections. Briefly, for this subcellular fraction analysis we isolated retinae from both eyes of n= 3 WT-P26, n=3 rd10-P17 and n=3 rd10-P26 mice and pooled them for further homogenization and lysate preparation steps in 3 (each group containing total of 6 retinae) groups as WT, rd10-P17 and rd10-P26 as shown in the Figure 3h WB.
Can the authors provide a loading control for western blot images in panel g and h?
Response: Included
The authors should include the names of the abbreviations from g and h panels in Figure 3 into figure legend.
Response: Included
Can the authors provide a quantitative analysis of the images presented in Figure 4? This would strengthen the authors’ conclusions.
Response: We thank the reviewer for this valuable suggestion. Figure 4 deals with the aggregation and cytoplasmic mislocalization of FET RBPs including FUS, HNRPN, TAF-15 etc. The semi-quantitative analysis of the immunoreactivity pattern is now provided in Table S2
Can the authors provide a loading control for western blot images in panel g?
Response: Coomassie blue as a loading control is included
Can the authors provide a quantitation of panel b?
Response: The semi-quantitative analysis for the immunoreactivity pattern is provided in Table S2
Can the authors provide a loading control for western blot images in panel c?
Response: Included
Reviewer 2
The article “Early Alterations of RNA Binding Protein (RBP) Homeostasis 2 and ER Stress-Mediated Autophagy Trigger Progressive Retinal Degeneration in the rd10 Mouse Model of Retinitis Pigmentosa (RP)” by Alfred Yamoah, Priyanka Tripathi, Haihong Guo, Leoni Scheive, Peter Walter, Sandra Johnen, Frank Müller, Joachim Weis and Anand Goswami, investigates changes in the expression of different proteins in the retina of rd10 mice and WT mice.
Authors hypothesized that RNA binding protein (RBP) aggregation and abnormal autophagy might serve as early pathogenic events, damaging non-photoreceptor retinal cell types. Authors have used combinations of immunohistochemistry (DAB, immunofluorescence), electron microscopy, subcellular fractionation, and Western blot analysis on the retinal samples from rd10 27 and wild-type mice The topic of this is of considerable interest, however, there are major concerns that need to be addressed.
Although the materials and methods section provide a lot of experimental details, it remains unclear how exactly the Authors divided the eyes for each assay, taking into account that fixation protocols for HE, immunocytochemistry and for WB and TEM are different.
How many sections Authors used for immunocytochemistry, how many for HE and TEM? Suppl. Table would be helpful to clarify this matter.
Response: We thank the Reviewer for presenting his/her general overview of the manuscript and highlighting our major findings. Even though we mentioned the sample sizes in the Methods as well as in the Figure legends sections, we agree that this aspect should be highlighted. In addition, we have included Table S3 clarifying all the necessary details on sampling.
2.4. Which type of hematoxylin was used for counterstaining in immunocytochemistry?
Response: We used H3136 hematoxylin from Merck
2.9. Transmission Electron Microscopy (TEM)
Response: Corrected
Did the Authors used only 2,5% glutaraldehyde without PFA for fixation? Also, did the Authors use any method for contrasting of ultra-thin section for TEM?
Response: We used fresh retinae and fixed them in 2.5% buffered glutaraldehyde for 48 hours. Ultrathin sections (70 nm) were mounted on grids, contrast-enhanced with uranyl acetate and lead citrate, and examined using a Philips CM10 transmission electron microscope equipped with a Morada digital camera
3.1 Increased Levels of Calcium-Binding/Buffering Chaperones and ER Stress in rd10 Retina
228 Before Noticeable Photoreceptor Loss at P17
In this section Authors present changes in the distribution, immunoreactivity and changes of
immunoreactivity of calcium-binding protein, PERK, PKC, GRP78/Bip and GADD -123 in WT
and rd10 retina as well as changes in expression of those proteins (WB). This subsection
contains a lot of information and is difficult to read, thus needs to be rewritten to be more
concise and understandable for the readers. Also, the heading of this subsection should be
modified to include both morphological and WB results.
Response: We regret the confusion, based on reviewers suggestions, we have now extensively revised the text for smooth flow and easier reading.
3.1 and 3.2 3.3 and 3.4. subsections are too descriptive. Authors should focus here on pure
results and present them in a concise manner. References and conclusions should go to the
discussion section.
Response: We extensively revised the text accordingly.
Discussion
Discussion should start with the main findings of this study and then discuss it with results of
other studies (similar results or opposite). Instead, at the beginning of the discussion there is a long sentence about how the retina is structured (what we can find in the textbooks). The conclusion is also missing at the end of the discussion. This section needs to be improved.
Response: The text was revised accordingly.
References:
- Chang, B., N. L. Hawes, M. T. Pardue, A. M. German, R. E. Hurd, M. T. Davisson, S. Nusinowitz, K. Rengarajan, A. P. Boyd, S. S. Sidney, M. J. Phillips, R. E. Stewart, R. Chaudhury, J. M. Nickerson, J. R. Heckenlively, and J. H. Boatright. "Two Mouse Retinal Degenerations Caused by Missense Mutations in the Beta-Subunit of Rod Cgmp Phosphodiesterase Gene." Vision Res 47, no. 5 (2007): 624-33.
- Gargini, C., E. Terzibasi, F. Mazzoni, and E. Strettoi. "Retinal Organization in the Retinal Degeneration 10 (Rd10) Mutant Mouse: A Morphological and Erg Study." J Comp Neurol 500, no. 2 (2007): 222-38.
- Roche, S. L., A. C. Wyse-Jackson, A. M. Byrne, A. M. Ruiz-Lopez, and T. G. Cotter. "Alterations to Retinal Architecture Prior to Photoreceptor Loss in a Mouse Model of Retinitis Pigmentosa." Int J Dev Biol 60, no. 4-6 (2016): 127-39.
- Newton, F., and R. Megaw. "Mechanisms of Photoreceptor Death in Retinitis Pigmentosa." Genes (Basel) 11, no. 10 (2020).
- Chang, B., N. L. Hawes, R. E. Hurd, M. T. Davisson, S. Nusinowitz, and J. R. Heckenlively. "Retinal Degeneration Mutants in the Mouse." Vision Res 42, no. 4 (2002): 517-25.
- Rosch, S., S. Johnen, F. Muller, C. Pfarrer, and P. Walter. "Correlations between Erg, Oct, and Anatomical Findings in the Rd10 Mouse." J Ophthalmol 2014 (2014): 874751.
.

Reviewer 2 Report
The authors describe using the rd10 mouse model of RP to investigate the association of RNA binding protein (RBP) aggregation and abnormal autophagy with early pathogenic events in the retina. The authors examine levels of various markers of autophagy, RBPs, FET proteins, and RNA stress granules using retinal samples. The authors show that these markers are increased before the onset of photoreceptor degeneration in the rd10 mouse model. Thus, the authors conclude that there is an association between RBP aggregation and abnormal autophagy with the photoreceptor degeneration in the rd10 mouse.
Although the data presented in this manuscript supports the authors’ claims, a major issue with this work is the lack of establishing a causative relationship between RBP aggregation and abnormal autophagy with the photoreceptor degeneration in the rd10 mouse. The authors rely on the lack of photoreceptor degeneration at P17 in the rd10 mouse to make their conclusions but there is a report that has found significant ERG deficits in these mice at P14, even though there is no photoreceptor degeneration at this age (PMID: 24683495). Thus, it is unknown whether the RBP aggregation and abnormal autophagy induces the photoreceptor degeneration in the rd10 mouse or occurs as a result from other pathogenic mechanisms that could explain the retinal phenotypes of these mice.
Other Minor Comments:
Introduction:
Lines 53-55: Cite references that support this sentence.
Lines 55-56: Cite references that support this sentence.
Lines 56-57: Cite references that support this sentence.
Line 73: Remove hyphen in dis-ease
Results:
3.1
The authors could measure the thickness of the ONL in Figure 1A to support conclusions.
The authors should include data on WT-P50 and rd10-P50 in Figure 1G since these age points are mentioned in the results.
The authors could combine the representative images and quantification of the images closer to each other in Figure 1 (for example, combine panels B and G, panel D and G, and panel E and H).
Did the authors verify the increased calreticulin staining with increased calreticulin protein on a Western blot?
The authors observed changes in SigR1 staining at both P17 and P26 ages but only in total SigR1 protein through Western blot analysis at P26. Can the authors explain these differences? Maybe this should be included in this section of the results?
Lines 269-272: The authors only found differences in GRP78 and PDI at P17 (not P16 as described in this paragraph) and not in PDI and pPERK. The authors need to revise this sentence accordingly.
Figure 1: The magnification of images is missing and should be added.
3.2
The authors do not include the quantification of their P62 western blot in Figure 2.
The authors should analyze the LC31 and LC3II bands separately and compare the amount of lapidated LC3II to LC3I since this ratio is more telling about the status of autophagy in the retinas of the mice than the overall protein amounts of LC3.
What are the magnifications of the images presented in Figure 2?
Can the authors provide and a quantitative analysis of the images presented in panel b of Figure 2? This would strengthen the authors’ conclusions.
3.3
Can the authors provide a quantitative analysis of the images presented in Figure 3? This would strengthen the authors’ conclusions.
Can authors clarify if the sample size in Figure 3h refers to number of retinas used in generating fractions or number of independent samples used for western blotting?
Can the authors provide a loading control for western blot images in panel g and h?
The authors should include the names of the abbreviations from g and h panels in Figure 3 into figure legend.
3.4
Can the authors provide a quantitative analysis of the images presented in Figure 4? This would strengthen the authors’ conclusions.
Can the authors provide a loading control for western blot images in panel g?
3.5
Can the authors provide a quantitation of panel b?
Can the authors provide a loading control for western blot images in panel c?
Author Response

(The authors gave the same response as above.)

Round 2
Reviewer 2 Report
The authors included compelling responses to this reviewer’s comments. Although it would have been nice to establish a causative relationship especially during the peer review process timeline, this reviewer can accept that the results in this paper can only support a correlative relationship between RBP aggregation, abnormal autophagy, and disruptions in calcium homeostasis with the retinal pathologies in the rd10 mouse. However, I would implore the authors to change ‘trigger’ in the title of their paper as it suggests a causative relationship between their findings and the phenotype of the rd10 mouse. I would also suggest to remove the length bar of the ONLs from the images since these distract from the retinal images in Figure 1, panel A. Other than that, this reviewer is satisfied with the changes made in this revision.
Author Response
March 13th, 2023
Dear Editors,
We are very grateful for the second round of the positive and constructive review of our manuscript entitled “Early alterations of RNA binding protein (RBP) homeostasis and ER stress-mediated autophagy triggers progressive retinal degeneration in the rd10 mouse model of retinitis pigmentosa (RP)”. please find below our point-by-point responses to the reviewers´ comments. The new version of our manuscript has been revised according to the reviewers´ suggestions (changes highlighted). We sincerely hope that you will find the revised version acceptable for publication.
Yours sincerely,,
Anand Goswami and Joachim Weis
Comments and Suggestions for Authors
Reviewer 1
Comment
The authors included compelling responses to this reviewer’s comments. Although it would have been nice to establish a causative relationship especially during the peer review process timeline, this reviewer can accept that the results in this paper can only support a correlative relationship between RBP aggregation, abnormal autophagy, and disruptions in calcium homeostasis with the retinal pathologies in the rd10 mouse. However, I would implore the authors to change ‘trigger’ in the title of their paper as it suggests a causative relationship between their findings and the phenotype of the rd10 mouse. I would also suggest to remove the length bar of the ONLs from the images since these distract from the retinal images in Figure 1, panel A. Other than that, this reviewer is satisfied with the changes made in this revision.
Response
We sincerely thank the reviewers for presenting his/her general overview of the manuscript and accepting our responses. We have now revised the manuscript based upon the above comments. We have replaced the word Trigger to Contribute- in the title and also removed the white scale bars.